# The Impact of Inorganic Systems and Photoactive Metal Compounds on Cytochrome P450 Enzymes and Metabolism: From Induction to Inhibition

**DOI:** 10.3390/biom14040441

**Published:** 2024-04-04

**Authors:** Dmytro Havrylyuk, David K. Heidary, Edith C. Glazer

**Affiliations:** Department of Chemistry, North Carolina State University, Raleigh, NC 27067, USA; dhavryl@ncsu.edu

**Keywords:** cytochrome P450, enzyme inhibitor, metal complex, photocage, photosensor, metabolism

## Abstract

While cytochrome P450 (CYP; P450) enzymes are commonly associated with the metabolism of organic xenobiotics and drugs or the biosynthesis of organic signaling molecules, they are also impacted by a variety of *inorganic* species. Metallic nanoparticles, clusters, ions, and complexes can alter CYP expression, modify enzyme interactions with reductase partners, and serve as direct inhibitors. This commonly overlooked topic is reviewed here, with an emphasis on understanding the structural and physiochemical basis for these interactions. Intriguingly, while both organometallic and coordination compounds can act as potent CYP inhibitors, there is little evidence for the metabolism of inorganic compounds by CYPs, suggesting a potential alternative approach to evading issues associated with rapid modification and elimination of medically useful compounds.

## 1. Introduction

Cytochrome P450 (CYP) enzymes are responsible for the biotransformation of xenobiotics and the biosynthesis of a variety of essential signaling molecules; as a result, their function is central to health. In humans, there are 57 different CYPs, with distribution in a wide variety of tissues, including the brain, skin, adrenal glands, kidneys, gastrointestinal tract, and liver [1,2]. Notably, liver CYPs are responsible for the metabolism of most drugs, with estimates as high as 75% of phase I dependent drug degradation [3] being performed by members of the CYP 1–3 families, which play the leading role in this process. Meanwhile, CYPs in various other tissues are involved in enzymatic processes to generate signaling molecules, such as hormones, and derivatives of arachidonic acid, which act as vaso-modulators, as well as autocrine and paracrine mediators [4,5]. Dysregulation of CYP activity due to alterations in transcription, single nucleotide polymorphisms (SNPs), subcellular localization, or environmental conditions, including the presence of inhibitors, can lead to aberrant metabolism. This, in turn, causes a variety of altered biological processes, including malignant transformation. Alternatively, the CYP metabolism of small molecules can lead to direct toxic effects through the creation of harmful entities that have local or systemic activities [6].

While there have been numerous studies on the impact of drugs, natural products, food components, and organic small molecule pollutants on CYPs, the activities of inorganic species on these enzymes are relatively underexplored. This topic requires a perspective that considers features of the diverse fields of inorganic chemistry, organic chemistry, medicinal chemistry, and enzymology. Notably, inorganic species can impact enzymes through a variety of mechanisms, as illustrated in Figure 1. In addition, they can also alter CYP expression profiles, as metal species are recognized as potential biological threats, and CYPs often serve in protective roles and are responsive to changing environmental conditions. 

We aim to provide here the first review of the modulation of CYPs by metals and metal compounds. It is important to note that each of the studies summarized here is limited to only a few target CYPs; as a result, they do not reflect on the impact of the investigated chemical systems across the diverse CYP family. As the types of experiments performed and the nature of the system under study can impact the interpretation of results, these are noted whenever possible, with information indicating if the experiments were performed with purified protein in vitro, in cells, or in vivo. Finally, a consistent challenge with inorganic systems is the fact that metal-containing agents are often, by nature, fluctional. For example, nanoparticle surfaces can be modified by biopolymers and small molecules, thereby changing their properties; some metal complexes can exhibit different oxidation states in the biological environment; conformational changes and ligand exchange can occur; and in some of the systems detailed here, energy and electron transfer takes place between the inorganic compound and the CYP. We comment on these features to alert the reader to possible confounding variables but also indicate cases when there is reason to treat the inorganic entities as stable and unreactive molecules. 

## 2. Metallic Nanoparticles, Metal Clusters, and Metal Ions 

There have been several studies on the impact of metal nanoparticles (NPs) on CYPs, and like other nanoparticle systems, such as those composed of silicon [7] or polystyrene [8], metallic NPs have been shown to alter CYP function and regulation. The potential for drug–drug interactions caused by unintended CYP inhibition by metallic NPs was reported by multiple groups, and this is a significant reason for concern for the medical application of any NP system. Notably, NPs are generally known to accumulate in the liver [9], though there are exceptions, including a study of orally dosed silver nanoparticles (AgNP) that were not found to deposit in the liver [10]. However, the biological effects of NPs are known to depend on their size, which impacts their partitioning from the vasculature into tissues, and their ability to transverse cellular membranes. Surface properties also play a very important role, and these can be impacted by synthesis conditions or by interactions with biopolymers and small molecules. This variability in NP characteristics results in elevated uncertainty as to the nature of the particles driving a phenotype and lower reproducibility; as a result, generalizations from one NP study to another should be viewed with caution.

In one report, AgNPs that are marketed as a supplement to provide immune system support were found to impact metabolic processes in a CYP-specific manner [11]. Using liver microsomes, the authors monitored the transformation of selected small molecules to determine the inhibitory effects on specific CYPs. It was reported that CYP2C9, CYP2C19, and CYP3A4 were most susceptible to inhibition, CYP2D6 and CYP2E1 exhibited moderate inhibition, and CYP1A2, CYP2A6, and CYP2B6 were the least impacted. The authors proposed a non-competitive inhibition mechanism, driven by interactions with the protein surface; this conclusion was based on the size of the NPs, which were assumed to exceed the capacity of the CYP binding pocket (however, the dimensions of the AgNPs in this report were not defined). 

Another study was performed that utilized molecular docking along with quantum mechanical (QM) analysis to evaluate the impact of Ag(I) ions and Ag_3_ clusters [12]. In this theoretical study, it was found that the Ag_3_ cluster interacted with amino acids in binding pockets of CYP2C9, CYP2C19, and CYP2D6 enzymes, but not CYP1A, CYP2E1, and CYP3A. This profile of selectivity for inhibition of CYP isoforms was somewhat consistent with the authors’ prior in vitro enzyme inhibition studies with AgNPs, though the proposed binding sites were different (i.e., the enzyme surface vs. active site). The difference in the proposed binding sites is related to the large discrepancy in size; the triatomic silver cluster is ca. 0.3 nm in diameter, at least 10-fold smaller than even the smallest traditional nanoparticle system, which is generally reported as 5 nm. 

Effects similar to those observed with AgNPs were found for 5–100 nm gold nanoparticles (AuNP) in studies with CYP2C9, CYP2C19, CYP2D6, and CYP3A4 [13,14]. The interaction of AuNP with CYPs could be a result of adhesion onto the enzymes, thereby obstructing the active site, or by blocking the interaction surface for binding to the reductase partner; alternatively, the NPs could cause destabilization of the enzyme or modulation of its structure or even induce disruption of the membrane structure around the CYP [14]. As P450s are integral membrane proteins, any nanoparticle system that has an affinity for membranes or disrupts the composition or structure of membranes can be anticipated to alter CYP activity.

Copper nanoparticles (CuNP) and microparticles (micro-copper) have also been studied in rats after oral exposure, and effects on both mRNA expression and activity of various CYPs were found in a dose-dependent manner [15]. The authors investigated CYP1A2, CYP2C11, CYP2D6, CYP2E1, and CYP3A4, and the effects were most pronounced for high-dose CuNP, with smaller effects observed with micro-copper. Another report found an impact of CuNP in vivo on rat brain CYP expression, which the authors attributed to increased oxidative stress and altered expression of nuclear receptors in the brain [16], while another study demonstrated an impact on CYP-mediated hormone metabolism [17]. In the latter investigation, 6.25 μg/mL of CuNPs downregulated the expression of CYP1A1, CYP1B1, and CYP3A4 but increased the protein expression of CYP11A1, CYP17A1, and CYP19; protein levels were altered by 2–3 fold, and estrogen levels were elevated by the same factor. The studies were performed both ex vivo, in isolated rat ovaries, and in vitro, in ovarian granulosa cells, with similar results demonstrating an increase in hormone production, accompanied by a decrease in CYPs involved in the breakdown of hormones and small molecules. Other studies with more complex nanoparticles, such as Mn_3_O_4_, have also revealed liver toxicity, reactive oxygen species (ROS) associated with CYPs, and altered CYP expression profiles [18]. Recent reviews have discussed this topic in some detail [19,20].

The impacts of mono- and divalent metal ions on CYPs have also been investigated, and these cations often appear to have opposite effects on enzyme activity. The most detailed studies were performed with purified proteins, where it was found that the alkali metals Na(I) and K(I) activate CYPs, inducing low- to high-spin shifts and increasing substrate binding affinity. This was rationalized in investigations of bacterial CYP101 (also known as P450_cam_) by a mutational and structural study of the enzyme that showed that K(I) engaged in interactions with the amino acids backbone carbonyls of E84, G93, E94, and Y96 in the BC loop and induced conformational changes [21]. Also investigated in P450_cam_, the divalent alkaline earth metals Ca(II) and Mg(II) had different effects, depending on concentration and incubation time. Initial titrations demonstrated an increase in the high-spin state and affinity for the substrate, with the following order of activity: no metal < Ca(II) ≈ Mg(II) < K(I). The metal ions were presumed to bind at an auxiliary metal binding site of the enzyme and slowly facilitated a conformation change to the inactive P420 form. However, this conformation was different from the previously reported P420 forms and was found to be reversible, in contrast to the irreversible nature regularly exhibited by the P420 species [22]. Moreover, divalent ions decreased the electron transfer activity from NADH in the presence of reductase partners putidaredoxin and putidaredoxin reductase, suggesting an alternative means of enzyme regulation [22].

Indeed, electron transfer from the reductase partner (generally cytochrome P450 oxidoreductase (CPR), also known as P450 oxidoreductase (POR)) is the rate-limiting step in P450 catalysis, and the reductase partner is usually present as the limiting reagent, making this a key potential bottleneck. One early study indicated that Cu(II), but not Ca(II), Co(II), Mg(II), Mn(II), or Zn(II), inhibited CYP reactions at micromolar concentrations in liver microsomes and with purified proteins [23]. The authors found that the copper exerted its effects through both interactions with CYPs and with POR, altering protein conformations and blocking electron transfer. However, other studies tested and eliminated interference with POR as a rationale for inhibition by metal ions; for example, Zn(II) was found to directly inhibit CYP3A4 with an IC_50_ value of 27 μM, and modulated its structure [24]. (It was also found to interfere with the activity of CYP3A4 mediated by *b_5_*, but this was found with 100 μM Zn(II), which is two orders of magnitude higher than anticipated concentrations of the free ion in tissues.) A recent report delved into this disputed subject [25]. In this study, the authors performed a computational study to investigate how ions and electric fields perturb CYP activity and determined that ions present in the active site may activate or inhibit CYPs, with effects observed over relatively long distances between the metal and the heme. Their results showed effects at distances up to 12–20 Å, which were dependent on the number and nature of the ions, potentially resolving the seeming conflict in the literature.

While divalent metals are generally inactivating, an anomaly among P450s is albaflavenone synthase (CYP170A1 from soil bacterium from *Streptomyces coelicolor* A3(s2)), which requires a divalent cation cofactor for its farnesene synthase activity [26]; the highest activity was found with Mg(II), Ca(II), and Mn(II). However, CYP170A1 is noted to be a product of unusual evolution, given its moonlighting activity as a novel terpene synthase. It was postulated that this chimerical enzyme is a product of either divergent or horizontal DNA transfer and contains the active site for a terpene synthase with the DDXXD and DTE signature sequences for divalent cation binding, along with the standard P450 fold and active site. Thus, the essential nature of the cation is not related to the P450 mechanism. 

Finally, while beyond the scope of this work, there are studies that have demonstrated that some metal oxides can have significant effects on CYPs. For example, arsenic, a common environmental pollutant, has been reported to alter CYP expression levels and activity [27]. In vivo investigations ranged from studies in rats chronically exposed to arsenic [28] to observations of acute effects in mice [29] to evaluations in human subjects with type 2 diabetes mellitus monitoring alterations in the pharmacokinetics of drugs as a function of arsenic levels [30]. The mechanism(s) for these effects are not understood.

## 3. Coordination Complexes and Organometallic Compounds

Cytochrome P450s are robust enzymes and have been proven to be amenable to direct interactions with *kinetically inert* metal complexes, which are defined as those that are not subject to ligand exchange or other structural rearrangements on the time scale of the experimental work. In contrast, labile metals such as Zn(II) or Cu(II) are expected to undergo rapid ligand exchange and thus may modify the protein in unanticipated ways through coordinative interactions with nucleophilic amino acids. Kinetically inert metal complexes include the Ru(II) and Ir(III) systems detailed below and are expected to retain their coordinated ligands, particularly when chelating ligands are used. This was illustrated in pioneering studies by Cheruzel and coworkers, which demonstrated covalent modification of surface amino acids through reactions with pendant reactive groups on the organic ligands of kinetically inert Ru(II) centers [31,32]. This allowed for the development of hybrid biocatalysts, where CYP enzyme activity was induced by photons [33,34]. These photocatalyst–enzyme chimeras were remarkably robust, achieving high turnover numbers and performing enantioselective catalysis [35]. Notably, the reactive functional groups on the ligands of the metal complexes could also be used to induce the construction of multi-protein arrays [36,37]. The diverse topics related to the interplay of metal complexes and CYPs were recently addressed in a comprehensive review [38].

### 3.1. Development of Inorganic Precursors for Photo-Controlled Inhibition of CYPs

Ru(II) complexes have been used for photocaging of P450 inhibitors, where a biologically active monodentate ligand is masked by coordination to a metal center and then released by light. The resulting products are the free CYP inhibitor and a Ru(II) complex that is assumed to contain an aqua ligand in place of the ejected inhibitor. This process allows for temporal and spatial control over enzyme activity and potentially overcomes concerns about promiscuous inhibition of the various biologically important CYPs that are present in different tissues. The approach has been applied with coordinating Type II inhibitors, where the engagement of the inhibitor with the metal compound was anticipated to prevent ligation of the iron heme. In this approach to photocaging, the Ru(II) complex itself is not anticipated to have any affinity for the CYP target and is not expected to undergo further ligand exchange reactions. However, premature release of the CYP inhibitor is a persistent challenge with these “caged” systems, even though they are developed from structures that are expected to be stable (inert) with regard to ligand exchange. This is likely due in part to experimental limitations, as exposure light is required to some degree for the handling and evaluation of the photocaged inhibitor, so some photochemistry will occur. More challenging to rationalize is the fact that monodentate ligands do indeed exchange, even when coordinated to second- and third-row metal centers, which are expected to only undergo ligand exchange on very slow time scales (with water exchange rate constants 10^−10^ < *k* < 10^−1^ s^−1^ [39]; these values decrease further with the engagement of strong field ligands, which increase the crystal field stabilization energy). However, ligand exchange rates can be higher in a complex biological environment, which contains a wide variety of nucleophiles that compete with the coordinated inhibitor. This topic is addressed in Section 3.5, below. 

The first report of photocaged P450 inhibitors was published by Glazer and coworkers in 2017, with a [Ru(bpy)_2_] (bpy = 2,2′-bipyridine) scaffold employed for caging metyrapone and etomidate [40], both of which are inhibitors of CYP11B1, also known as steroid 11-beta-monooxygenase. In addition, a new P450 inhibitor was created that represented a hybrid of the two drugs (the pink monodentate ligand in **1**, Figure 2). An engineered variant of the bacterial enzyme CYP102A1 (P450_BM3_) was chosen for the studies. While wild-type P450_BM3_ is selective for fatty acid, amide, and alcohol substrates [41], bioengineering efforts produced mutants that are able to bind and modify a variety of nonnative substrates, including a number of drugs [42,43,44,45]. This makes specific BM3 mutants convenient models of promiscuous drug-metabolizing hepatic CYPs, and in contrast to human and other mammalian CYPs that are membrane-bound, BM3 is soluble, relatively stable, and readily generated in high yield, facilitating structural and biophysical studies. The enzyme was truncated to remove the C-terminal reductase domain that is unique to BM3 (other CYPs rely on protein:protein interactions with reductase partners) [46]. The truncation removed the flavin binding domain, eliminating the potential confounding effects of irradiation on the photoreactive FAD and FMN cofactors. To support the studies with BM3, additional experiments were performed in pooled human liver microsomes (HLMs) to evaluate inhibitory efficacy on the various CYPs responsible for first-pass metabolism that metabolize resorufin ethyl ether.

All three Ru(II) complexes were studied in the dark, and notably, all complexes exhibited enzyme inhibition without irradiation at the micromolar level (IC_50_ = 0.6–6.8 µM). A 15- and 18-fold enhancement in potency was observed upon irradiation (λ_irr_ = 450 nm) for complexes with etomidate and metyrapone, and the novel P450 inhibitor (**1**) had an IC_50_ value of 0.05 μM after irradiation with light, providing a 136-fold difference between activity in the dark and the light (Table 1). While metyrapone contains two coordinating groups, hypothetically allowing for simultaneous engagement with both Ru(II) and the iron heme, this was not possible for etomidate and the inhibitor in **1**, indicating that binding of the complex must occur through a different orientation of the inhibitor group within the active site. Like the experiments in BM3, the studies in HLMs demonstrated that **1** (100 μM) induced a ~10% reduction in resorufin ethyl ether turnover in the dark, and this increased to complete inhibition upon activation with light (λ_irr_ = 450 nm). Notably, the inhibition observed with light activation was greater than that for treatment with a 100 μM concentration of the organic inhibitor alone, suggesting the release of more than one equivalent of the inhibitor.

The seeming advantage of this increased potency upon photo-uncaging reflects a particular challenge in using the [Ru(bpy)_2_] and other *bis*-bidentate complexes as cages: either one or two monodentate ligands could be released, and the photochemical quantum yields for the first and second ligand are highly variable. Alternatively, the combination of tridentate ligands and strain-inducing bidentate ligands creates stoichiometric photocages that have more predictable photochemistry. The [Ru(tpy)(NN)] scaffold, where tpy is the tridentate 2,2′;6′,2”-terpyridine ligand, and NN are bidentate ligands, has been used for the caging of abiraterone, a human CYP17A1 inhibitor approved for the treatment of prostate cancer (shown as the lavender monodentate ligand in **2**, Figure 2) [47]. This [Ru(tpy)(NN)] scaffold was also combined with a novel inhibitor of CYP1B1 (the green stilbene system in **3**) [48], and inhibitors of CYP3A4 (the blue pyridyl system in **4**) [49]. Abiraterone was caged by the [Ru(tpy)(dmbpy)] (dmbpy = 6,6′-dimethylbipyridine) center and its binding affinity for CYP17A1 and cytotoxicity against DU145 prostate cancer cell line were evaluated in the dark and after activation with light (λ_irr_ ≥ 395 nm). Titration of CYP17A1 with increasing concentrations (0–280 nM) showed that complex **2** bound in the dark, although this produced a different spectral profile from the binding interaction observed after irradiation or with abiraterone, indicating a different binding mode. Significant differences in the cell viability in the light vs. dark were observed at ~1000-fold higher concentrations (40–100 µM) compared to binding to the enzyme K_d_ (90 nM) [47]. 

Utilizing the same synthetic strategy, Kodanko, Turro, and colleagues aimed to develop photocaged inhibitors of the drug-metabolizing hepatic CYP3A4. The inhibitors were based on two pyridyl-substituted analogs of ritonavir, an antiviral protease inhibitor that is also known to inhibit 3A4. The series of Ru(II) complexes was expanded by the application of Me_2_dppn as a strained bidentate ligand (Me_2_dppn = 3,6-dimethylbenzo[i]dipyrido [3,2-a:2′,3′-c]phenazine). All intact complexes exhibited inhibition of CYP3A4 at sub-micromolar concentrations without light activation. Moreover, coordination of the organic CYP3A4 inhibitor (the blue structure in complex **4**; IC_50_ = 1.54 µM) produced a Ru-based inhibitor with a 6-fold superior potency (IC_50_ = 0.25 µM) than the parent inhibitor. Consequently, the photoactivation of these compounds had a negligible effect on CYP3A4 inhibition. The Ru(II) complexes with the Me_2_dppn co-ligand combined photodissociation with efficient ^1^O_2_ production and one compound was tested in combination with vinblastine against DU-145 cells, providing a synergistic effect [49].

Inclusion of the biq (biq = 2,2′-biquinoline) ligand in Ru(II) photocages can be used to shift the absorption profile to longer wavelengths due to the lower energy of the metal to ligand charge transfer (MLCT) transitions, which depend on the lowest unoccupied molecular orbital (LUMO) of the conjugated biquinoline ligand [50]. This allows for the activation of [Ru(tpy)(biq)] photocages with longer wavelengths of light [51], which is advantageous for the ability to achieve activation with greater depths of light penetration into tissues. The [Ru(tpy)(biq)] scaffold was used for the development of photoactive inhibitors of CYP1B1, an extrahepatic CYP associated with malignant transformation and chemotherapeutic inactivation. The photocaged compound exhibited a dark IC_50_ value of 0.19 µM and was a 16-fold more potent inhibitor (IC_50_ = 0.012 µM) upon irradiation with red light (λ_irr_ = 660 nm). In an effort to preserve the favorable photophysics but eliminate some of the issues with [Ru(tpy)(biq)], additional optimization of the Ru(tpy) scaffold was undertaken. The incorporation of carboxylic acids into the biquinoline, using the [2,2′-biquinoline]-4,4′-dicarboxylic acid ligand (bca), reduced cellular toxicity caused by the metal complex such that no adverse effects were observed up to 100 μM concentrations [51]. This scaffold, shown in compound **3** (Figure 2), was then used for photocaging of a novel and potent CYP1B1 inhibitor (IC_50_ = 310 pM), which resulted in the precise control over enzyme activity with red light. The key features that enable photoactivity index (PI) values of >6300 were multifaceted and included the sub-nanomolar potency of CYP1B1 inhibitor, the high purity and thermal stability of the intact Ru (II) complex (>98%), and the high degree of photosubstitution upon activation with red light (>70% of photorelease) [48].

### 3.2. Binding of Metal-Containing “Wires” with P450 Proteins

Metal complexes tethered to substrates and other “bait” CYP-binding groups were developed as photosensors for different CYPs to facilitate energy and electron transfer investigations. The seminal work in this area was performed by Gray, Winkler, and collaborators. The initial studies demonstrated optical detection of cytochrome P450_cam_, where a [Ru(bpy)_3_] type scaffold was linked to substrates or ligands (ethylbenzene, adamantane, and imidazole) [52,53]. Dissociation constants were examined for the different Ru(II) probes based on emission quenching by P450_cam_, providing K_d_ values of 0.7–6.5 μM. The distance between the Ru(II) and iron heme was also determined to be ~ 20 Å by Förster energy transfer kinetics and the native substrate, camphor, could competitively displace the Ru-substrate wire, restoring emission. Detailed structural studies were performed, as discussed in Section 3.6 below.

Additional investigations were performed by Gray, Winkler, Goodin, and colleagues using this “wire” or “tethered substrate”–photosensitizer approach with the closely related enzyme nitric oxide synthase (NOS), which closely resembles CYPs [54]. (Notably, nitric oxide produced by NOS under inflammatory conditions also plays a role in CYP regulation [55]). Metal-conjugated substrates based on the amino acid arginine were developed to target the active site containing pendant Re(I) [56,57] and Ru(II) [17] diimine complexes. In addition, the first cofactor-tethered system was created based on a Ru(II) polypyridine complex conjugated to the redox-active tetrahydrobiopterin [58]. This left the enzyme active site open, allowing for the binding of substrates for chemical transformation and direct observation of photoinduced reduction and binding of carbon monoxide to the iron heme. In several of these and other studies, the photoexcited metal center could be competitively displaced by the natural binder or synthetic inhibitors, providing light-up probes for inhibitors. Photoinduced electron transfer from the metal centers to the heme allowed for the direct creation of reactive heme redox states [57,59,60].

Recently, Kodanko and others developed Ru(II) and Ir(III) based photosensors tethered to pyridine for monitoring the occupancy of the CYP3A4 active site [61,62]. CYP3A4 is a promiscuous enzyme with a large volume active site cavity; the size depends on the nature of the substrate or inhibitor bound within the enzyme, and volumes up to 1400 Å^3^ have been reported [63,64,65]. In contrast to prior CYP and NOS systems studied, even the octahedral Ru(II) and Ir(III) complexes are able to bind within the active site. The authors compared the binding and inhibitory activity of the photosensors with the different metal centers and concluded that the monocationic Ir(III) was significantly preferred over dicationic Ru(II) complexes. The closest structural analogs containing these two metals exhibited a difference of >300-fold, with K_d_ values of 170 nM and 53 μM; the best iridium inhibitor in this series exhibited a K_d_ value of 70 nM. Complex **5** exhibited low toxicity in CYP3A4-overexpressing HepG2 cells (EC_50_ > 50 μM) and inhibited CYP3A4 at nanomolar concentrations [61]. Further optimization of the Ir(III) photosensors was focused on the modulation of different co-ligands, shortening the length of the linker between the metal center and the heme-ligating pyridine, and isomerism of the pyridine ligand (*meta* vs. *para*-substituted). It was concluded that the hydrophobic and aromatic interactions mediated by the Ir(III) scaffold primarily define the binding ability and inhibitory activity. The optimized CYP3A4 probe (complex **6**) possessed a 3-fold higher binding affinity, with a K_d_ value of 24 nM; however, it was cytotoxic against HepG2 cells at the micromolar range (EC_50_ = 16 μM) [62]. Unfortunately, the lipophilic efficiency of the new metal-based inhibitors was not discussed in these reports, but it seems likely that the lower charge of the iridium complex was more compatible with the hydrophobic binding site.

It should be noted that various organic systems have been studied with characteristics similar to these “wires”. For example, organic fluorophores have been tethered to binding groups for P450_cam_ [66,67,68] and CYP1B1 [69], including a system with a pendant metal chelating group [70].

### 3.3. Photostable Inorganic Compounds Bearing Coordinated CYP Inhibitors

A variety of CYP inhibitors have been coordinated to metal centers and organometallic compounds; a selection is shown in Figure 3. These include letrozole, an inhibitor of aromatase (also known as CYP19A1), which was coordinated to Cu(I) [71], Cu(II), Ni(II), or Co(II) [72], and Ru(II) [73]. Castonguay found that ruthenium arene compounds containing another aromatase inhibitor, anastrozole, were more effective than those with letrozole, with studies that included CYP19A1 inhibition, in vitro cell cytotoxicity, and in vivo evaluation, which demonstrated an appealing lack of toxicity in zebrafish embryos. Several of the compounds exhibited the same potency as anastrozole, but some were labile, and for those, it is not possible to determine what fraction of the compound remained intact during the enzyme activity studies. However, the most stable system, compound **7** (Figure 3), still acted as an inhibitor. The level of inhibition was significantly higher than the concentration of free anastrozole that would be expected from the stability studies, and thus, a supplementary contribution from the intact complex was suggested [74]. Moreover, an analogous coordination isomer **8**, which contains the anastrozole bound via a nitrile group rather than the triazole ring, was found to be highly cytotoxic [75]. Unfortunately, due to this feature, complex **8** was only tested for aromatase inhibition in low concentrations (1–100 nM), and no significant activity was observed for the metal complex, whereas anastrozole was active at > 10 nM. These studies and others on rationally designed compounds for breast cancer treatment were recently covered in a detailed review [76].

A prodrug strategy was considered with anastrozole coordinated to vitamin B12 through a transplatin group, as shown in compound **9**. The B12 was incorporated to act as a targeting group for cancer cells, and the compound was designed with the metal centers in order to be activated by reduction. Unfortunately, CYP inhibition was not evaluated for this system [77]. Ketoconazole, an imidazole containing antifungal agent that inhibits the fungal cytochrome P450 enzyme 14α-demethylase (CYP51A1) has also been coordinated to a number of metal centers [78,79,80,81,82,83,84], and many of the systems exhibited promising biological activities as anti-infective agents. Unfortunately, there were no studies of direct enzyme inhibition.

Fluconazole, a first-generation triazole drug that targets fungal CYP51A1, has also been transformed into an inorganic system [85]. Ferrocene derivatives such as **10** were synthesized and characterized both in vitro and in vivo in a mouse model of *Candida* infection. The metal-containing compound demonstrated efficacy superior to the parent drug; it was 400-fold more potent in clinical isolates and produced improved histology scores in vivo. The mechanism of action via inhibition of fungal CYP51A1 was maintained, but additional benefits included activity in resistant fungal strains and a pro-inflammatory cytokine response. This was accompanied, however, by a promiscuous inhibition profile through a panel of liver CYPs, with a 4–150-fold higher level of activity than the parent drug fluconazole. Thus, the inorganic system was more potent across several CYPs, not just the anticipated target. These results suggest that the metal fragment plays a significant role in CYP binding and inhibition.

### 3.4. CYP Inhibition by Metal-Containing Compounds

Some metal complexes that do not contain coordinated CYP inhibitors have been found to exhibit enzyme inhibitory activity. For example, the coordination complexes [Cu(4,7-dimethyl-1,10-phenanthroline)(acetylacetonate)]^+^ and [Cu(4,7-dimethyl-1,10-phenanthroline)(H_2_O)_2_]^2+^ were found to be irreversible, competitive inhibitors of CYP1A1 (IC_50_ = 7.5 μM; IC_50_ = 3.5 μM) [86]. Inhibition was also observed for CYP1A2, 2B1, and 2B2. Homoleptic Cu(II) complexes of 8-hydroxyquinoline (HQ) and heteroleptic complex **11** containing one HQ and one uracil derivative [87] inhibited CYP19A1 with IC_50_ values of 1700 and 300 nM (compared to letrozole, with an IC_50_ of 330 nM in this study). While the compounds were cytotoxic, the enzyme activity assay was not performed in live cells, so the inhibition results cannot be attributed to cell death. Other copper complexes containing one 5-methyl-1,10 phenanthroline ligand and one quinolinato ligand but differing in the counterion (NO_3_^−^ vs. BF_4_^−^) were characterized for their inhibitory effects in human liver microsomes on CYP3A4/5 (IC_50_ = 2.46 and 4.88 μM), CYP2C9 (IC_50_ = 16.34 and 37.25 μM), and CYP2C19 (IC_50_ = 61.21 and 77.07 μM) [88]. These effects were attributed to a non-competitive type of inhibition. Optical binding studies were performed and an apparent Type I shift was observed, with K_s_= 7.59 ± 1.80 μM and 8.56 ± 1.00 μM (NO_3_^−^ vs. BF_4_^−^). The data for the metal compounds are consistent with direct binding, but notably, the ligands had no apparent binding to CYPs, as determined by optical spectroscopy. (It must be noted that polyaromatic hydrocarbons are substrates of several CYPs, so it is possible that the ligands do bind within the active site.) Moreover, titration with Cu(NO_3_)_2_·3H_2_O gave a similar optical signature in the same concentration range (K_s_= 9.81 ± 0.6 μM), suggesting that the copper ion, not the intact copper complex, is responsible for binding and inhibiting the CYPs. As the stability of the complexes was only moderate (approximately 8% degradation was observed in plasma over 120 min) and copper is considered to be kinetically labile, it is reasonable to consider ligand exchange that culminates in the release of the metal ion, which may then bind to the enzyme. It is possible that the copper interacts in a manner similar to that of Ag^+^ ions and clusters discussed earlier, which were also potent against CYP2C9 and 2C19 (although not 3A4).

Inhibition studies were also performed with the clinically utilized inorganic platin agents, cisplatin, carboplatin, and oxaliplatin, along with three other platinum compounds, in CYP1A2, CYP2A6, CYP2B6, CYP2C8, CYP2C9, CYP2C19, CYP2D6, CYP2E1, and CYP3A4 [89]. None of the approved drugs were significant inhibitors. However, transplatin did have a noteworthy effect, and transplatin is known to undergo more rapid ligand exchange than the *cis* platinum analogs.

### 3.5. Impact of Stability of Metal-Containing Compounds on CYP Inhibition

As discussed above, ligand exchange can result in the metal center itself acting upon the enzyme or releasing a ligand that is an inhibitor. Indeed, the development of selective, stable, and responsive agents with sufficient photoactivity indices (PI > 100) for photocaging applications is quite challenging, and this is likely due in part to this feature of ligand lability. Surprisingly, low PI values are a persistent problem for Ru(II) photocages, with dark IC_50_ values commonly being in the μM range. For example, PI values are generally below 40 for caged inhibitors of cysteine proteases [90,91,92], nicotinamide phosphoribosyltransferase (NAMPT) [93], and tubulin polymerization [94]. The instability of the intact complexes in biological media, where the inhibitors are released over time, can be one of the reasons for these observations. The stability of each compound is dependent upon the specific inhibitor and metal center structures; for example, the Ru(II) complex developed by Glazer, Heidary, and Havrylyuk containing a pyrimidyl-based CYP1B1 inhibitor exhibited thermal ligand release (57%) and dark protein inhibition at sub-micromolar concentrations, resulting in the low PI value of 16. In contrast, the pyridine analogs were far more stable [48]. The same trend has been observed by Castonguay and coworkers for organoruthenium(II) complexes bearing anastrozole. Complexes that exhibited low stability (>42% of ligand release after 1.5 h of incubation) possessed the same potency for aromatase inhibition as anastrozole. The highly stable complex **7** (which only released 4% of the ligand) was less potent in the CYP19A1 inhibition assay [74].

### 3.6. Structural Studies to Elucidate the Interactions between Inorganic Compounds and CYPs

Both crystallographic and computational approaches have been utilized to attempt to understand how metal compounds engage with CYPs. Notably, in most of these studies, the metal center is employed to protect the coordinating group of an inhibitor. This approach stands in marked contrast to the seminal work by the Meggers research group on inorganic enzyme inhibitors [95], which demonstrated that organometallic compounds could be used to mimic the structure of the kinase inhibitor, staurosporine. In Meggers’s design, the metal is used as a scaffold framework to orient the organic components, not to prevent a particular interaction between the organic system and the enzyme target. This runs counter to the design for most of the CYP-targeting inhibitors, where Type II inhibitors are coordinated to the metal center in a coordination complex or organometallic compound, thereby preventing the inhibitor from achieving the preferred binding mode with the iron heme. This was originally expected to block binding to the CYP, but the plasticity of CYP binding sites compensates with alternative binding modes.

This plasticity is illustrated by studies of a Ru(II) complex coordinated to a ritonavir analog, where the intact complex was found to be more potent than the original organic inhibitor [49]. The result is surprising not only because the Type II binding is prevented by the Ru(II) complex, but also due to the large differences in size and properties of the parent organic and derived metal-containing inhibitors, which would be anticipated to significantly change affinity. Figure 4 compares the crystal structure of CYP3A4 with ritonavir to that of the metal complex and shows how the binding pocket can accommodate these large compounds through an almost 180-degree rotation about the x-axis and a 180-degree rotation about the y-axis of the ritonavir analog vs. the metal complex. The structures agree with optical spectroscopy that indicated no nitrogen coordination to the heme for the ruthenium complex [49].

In order to determine similarities and differences in CYP as a function of binding the organic inhibitor vs. the metal-containing inhibitor, we analyzed PDB 4D78 and 7KS8 in detail. While both protein structures are similar and the backbones overlay, parts of the structure are disordered in the CYP3A4 ruthenium complex, as evidenced by the fact that the residues between positions 209 and 215 and between positions 263 and 288 could not be refined. In contrast, the CYP3A4 ritonavir structure had only two residues (265 and 266) that were unresolved. Contacts that are within a distance of 4 Å from the ritonavir analog are the heme, R105, F108, S119, I120, L211, F304, A305, T309, I369, and L482. In contrast, the amino acid contacts for the ritonavir analog for the ruthenium complex do not involve F108, I120, L211, and F304, but make new contacts with I301, A370, M371, and G481. The common contacts were R105, S119, A305, T309, I369, and L482. The binding for the ruthenium complex seems to rely on hydrophobic contacts around the ritonavir analog, and contacts with dimethyl-bipyridine and terpyridine rely on hydrophobic contacts as well as the π–π interaction with F57, which may allow for the higher binding affinity observed for the ruthenium complex.

These structural studies are interesting to consider in the context of the extensive literature on structure–activity relationships in CYPs based on probing amino acid active site interactions via mutagenesis, and how modulating individual residues alters the structure and catalytic selectivity. For example, studies on CYP2D6 revealed eight active site residues that play important roles in substrate binding and selectivity [96]; similar work was performed on 2C19 where mutations of three amino acids induced binding of a non-native substrate [97]. Studies on CYP2B1 demonstrated alterations in the metabolism of progesterone and chloramphenicol as a result of mutation [98]. Computational investigations of CYP2A6 variants with 8-methoxypsoralen demonstrated 3-fold changes in K_m_ values and 12-fold changes in IC_50_ values for four mutants [99]. Interestingly, docking studies with one variant, R203S, revealed a 180-degree rotation of the substrate, similar to what was observed with the metal-bound ritonavir analog. A number of studies on CYP3A4 demonstrated the impact of active site mutation on structure and plasticity [100], as well as activity and selectivity [101], and multiple binding sites were reported in the active site [102,103]. More extensive mutations were investigated with the creation of a cysteine-depleted CYP3A4, which exhibited alterations in structure and flexibility but maintained the ritonavir binding orientation observed in the wild-type enzyme while increasing activity towards a fluorescent substrate; this study also highlighted that crystal packing can affect the ligand binding mode [104].

While these reports illustrate that large changes in the active site have been characterized for many highly promiscuous CYPs, other structural studies have revealed only minor changes in protein conformation and contacts in P450s when bound to disparate substrates. For example, a detailed biophysical investigation compared the wild-type (and thus selective) P450_BM3_ to a pentuple mutant [105] that was promiscuous and capable of binding to a number of chemically disparate substrates [106]. This study provided information for two CYPs that exhibited radically different degrees of promiscuity, but the highest possible sequence homology, as there were only five amino acids that differed in the two sequences. Notably, there were significant alterations in the stability of the two protein forms and differences in how the binding of small molecules such as fatty acids vs. Type II inhibitors perturbed the stability. However, the crystal structures of the different forms of the protein bound to palmitic acid vs. metyrapone were remarkably similar [105]. This study highlighted the limitations in using crystal structures to fully understand how CYPs engage structurally diverse substrates or inhibitors. Conformational capture is one explanation for why crystal structures of CYPs do not reflect multiple, diverse conformations and binding modes that may exist in solution [68]; this will be the case if higher energy conformations do not crystalize. This can be a result of tight binding vs. weakly binding interactions, where the enzyme samples multiple conformations only in the latter case [107].

The binding of “wire” sensitizer-linked substrates and inhibitors is easier to rationalize than the binding of large metal complexes because in the wire systems, the binding group is only modified with a linker rather than a metal center. The first example of this is the crystal structure of the P450cam:Ru-C9-Ad complex (PDB 1QMQ) [52]. The structure demonstrated that the adamantane moiety resided at the P450 active site, while the methylene linker occupied and defined a large channel from the enzyme surface to the heme. The [Ru(bpy)_3_] docked at the surface of the protein, with no significant distortion of the nearby residues. Notably, the structures showed the protein in an open conformation, and the linker helped to visualize the substrate access channel.

Analogous to structural studies on these early sensors, crystal structures were solved for CYP3A4 with a pyridyl group tethered to an iridium complex [61]. The binding was drastically more potent with iridium than ruthenium, particularly for a complex with two phenylquinoline co-ligands (K_d_ = 70 nM) compared to ruthenium complexes with bpy or phen co-ligands (K_d_ = 53 μM and 23 μM). The larger hydrophobic surface area and lower overall charge on the iridium complex could be driving this 2–3 order of magnitude higher affinity. This interpretation is supported by the crystal structure (PDB 7UAY), which indicates that hydrophobic contacts are made for the co-ligands, with F108 being common with the ritonavir analog. For the phenylquinoline, the phenyl ring contacts F108 while the quinoline has contacts with F220 and L221. R105, S119, A305, and T309 make contact with pyridine and the linker, which also occurs with the ritonavir analog. S119 contact with the sulfur in the linker, along with R105 interaction with the carbonyl in the linker, is not seen with the ritonavir analog.

Rather surprisingly, the inhibitors are somewhat selective (28- and >36-fold) for CYP3A4 compared to CYP2C9 and CYP1A2, indicating that the combination of ligand and complex both play roles in binding [61]. It was also noted that the active sites of these other CYPs are much smaller (375 and 470 Å^3^ vs. values up to 1400 Å^3^ for 3A4) [63,64,65], so the large inhibitor size may be a key factor. Several structures solved with 10 different iridium complexes with CYP3A4 suggest that hydrogen bonding between the inhibitory ligand and S119 is important for potency.

Docking simulations were used by Castonguay to support the hypothesis of an interaction between the aromatase protein and the inhibitor-containing ruthenium complex **7** [74]. The in silico binding was dependent on the structure of the inhibitor, as the next-generation organoruthenium complex with anastrozole, **8**, did not have the same effect. In this case, the simulation showed that the interaction between complex **8** and the enzyme was energetically unfavorable [75]. However, the highly stable system was characterized by cytotoxicity in the sub-micromolar range (EC_50_ = 0.3–0.6 μM) against various cancer cell lines.

In our experience, most docking programs have difficulties with metal compounds, and the predictions are highly dependent on the crystal structure used in the modeling, as open or large active sites are better able to accommodate the metal centers. Recently, Bonnet and colleagues reported the first open-source program for docking inorganic systems, MetalDock [108]. This program allows for simulations of reactive metal systems that will engage with amino acids, or binding predictions for inert compounds where the metal can be considered a hypervalent carbon atom.

### 3.7. Metabolism of Inorganic Drugs by Liver CYPs

Data in the literature on the metabolism of metal compounds by liver CYPs is sparse. However, information was culled from reports of the FDA-approved metallodrugs auranofin and the platin agents. Auranofin ([2,3,4,6-tetra-o-acetyl-L-thio-β-D-glycopyranp-sato-S-(triethyl-phosphine)-gold]) has been used for the treatment of rheumatoid arthritis and has been considered for repurposing for a variety of other medical applications. This drug has a well-understood safety profile, and while there are no studies we could find in the literature discussing metabolism by CYPs, the extraordinarily long plasma half-life (17 days [109]) argues against CYP modifications, as these should increase solubility and the elimination rate. Cisplatin, which is used extensively in the clinic as a chemotherapeutic, is not reported to be metabolized by P450s, but there is a correlation between the toxic effects of cisplatin and CYP450 enzymes, as discussed in a recent mini-review [110]. The platin agents are considered to be subject primarily to renal clearance, and the understanding of their metabolism by CYPs and their impact on CYP metabolism may likely be incomplete.

There have been a few studies on the metabolism of metal compounds in development, and most have indicated that the compounds are potentially poor substrates. Copper complexes were investigated with human liver microsomes, and 70–78% remained after one hour. This suggests a moderate metabolic rate but must be considered in the context of the intrinsic stability of the complexes where ca. 6% was found to have degraded over 30 min in plasma [88]. A recent report provided a tutorial on investigations of the stability of organometallic systems, including metabolism by CYPs in liver microsomes [111].

### 3.8. Induction of CYPs by Metals, Metal-Containing Compounds and Drugs, and Enhanced Toxicity

The expression of CYPs is commonly induced by a wide variety of materials and drugs. The mechanism can involve direct activation of transcription (usually through binding to a variety of transcription factors) [112,113], but it can also be a downstream consequence of cellular responses to the compound or drug’s mechanism(s) of action. Nanoparticles have been shown to induce or alter CYP expression, with studies on CuNP [15], AuNP [13,114,115,116], AgNP [117], Mn_3_O_4_ NPs [18], and others; see the discussion above in Section 2. This effect is not limited to inorganic NPs or even solid materials, as injection of common “soft” materials such as polyethylene glycol-coated liposomes (PEG-L) has also been associated with upregulation of CYP expression [118].

Heavy metals like Hg(II), Pb(II), and Cu(II) induced CYP1A1 expression through the binding of the aryl hydrocarbon receptor/xenobiotic-responsive element (AhR/XRE) [119]. Another study showed that As(III), Cd(II), and Cr(VI) also increase CYP1A1 mRNA levels at the transcriptional level via AhR, but decrease enzyme activity at the protein level [120]. Alternatively, there is regulation of CYPs by the NF-E2-related factor 2 (Nrf2) [121], which is a regulator of cytoprotective enzymes involved in the response to toxins. This was demonstrated in a study that showed that cadmium chloride induced CYP2A5 mRNA and protein levels in Nrf2 +/+ mice but not in the Nrf2−/− mice [122].

While activation of transcription factors is well documented, an alternative mechanism would be the engagement of a metal response element (MRE). An MRE was found in the CYP1A1 promoter region in fish, which would provide a direct regulatory role for metals in vertebrate CYP expression [123]. An alternative, and indirect, process is observed with the metal-based drug, cisplatin, which has been found to induce nephrotoxicity in a manner mediated by CYP2E1 [110]; this is ameliorated in CYP2E1 knockout mice [124]. One plausible mechanism to explain this is that cisplatin interacts with the CYP2E1 protein, causing futile cycling and H_2_O_2_ formation; eventually, the protein is oxidatively damaged, causing the release of heme and iron. The free iron then is involved in the generation of additional ROS. Alternatively, the natural product limonin has been shown to protect against cisplatin-induced acute kidney injury through inhibition of CYP3A4 [125]. BOLD-100, an investigational new ruthenium drug in Phase II clinical trials, also exhibits effects on CYPs. Both mRNA and protein levels of CYP1B1 are elevated with BOLD-100 treatment in MCF7 breast cancer cells [126]; notably, CYP1B1 induction is associated with chemotherapy resistance. Another study demonstrated that BOLD-100 caused alterations in gene expression for a variety of CYPs involved in lipid and steroid metabolism [127]. These studies highlight the various ways that inorganic systems can alter CYP expression levels and emphasize the need to thoroughly investigate the impact of metal-containing agents intended for medical purposes.

## 4. Discussion and Conclusions

While CYPs are presumed to act primarily on hydrophobic organic substrates, inorganic ions, clusters, particles, and complexes (both organometallic and coordination complexes) can also have a significant impact on CYP activity. Perhaps this should not be unanticipated, given the number of ways enzymes engage with diverse molecular entities, but what is surprising to us is the capacity for *selective* engagement and inhibition of only particular CYPs, especially when a preferred mode of binding is not possible. Our detailed analysis and comparison of the inhibitory activity and cytotoxicity for different metal-based P450 agents revealed the following observations. (1) The majority of metal complexes with pendant CYP inhibitors exhibit inhibition of the protein of interest despite their purpose of utilization (prodrugs, cytotoxic agents, or photosensors); (2) two explanations for their activity are the release of inhibitor under biological conditions (for example, anastrozole derivatives) or the direct binding to P450 protein (as with CYP3A4 photosensors); (3) the development of photoactive metal-based prodrugs has been more successful for CYPs with a smaller volume of the active site, and the photocontrol of enzyme inhibition for promiscuous CYP3A4 is challenging due to the flexibility of the protein that allows encapsulation of the intact complexes within the active site (Figure 4); (4) while a model Ru(II) complex (with a pyridine coordinated to the metal center instead of the organic inhibitor) did not inhibit CYP1B1 activity in concentrations up to 30 μM [48], a model monocationic Ir(III) complex exhibited inhibition of the promiscuous CYP3A4 at sub- and single-micromolar concentrations (IC_50_ = 1 μM), reinforcing the fact that each metal-containing compound may have unique behavior, and this may be correlated to lipophilic efficiency; (5) metal-containing systems may have altered and augmented activities, as exemplified by the attachment of ferrocene to fluconazole (compound **10**, Figure 3), which increased activity against the target CYP while also adding new and potentially useful biological interactions. A related approach is represented by the coordination of aromatase inhibitors to the organoruthenium scaffolds, which resulted in cytotoxic compounds at sub-micromolar concentrations (Compounds **7** and **8**, Figure 3). The impact of aromatase inhibition on cytotoxicity is unclear, and it is possible that cytotoxicity is driven by the intact complex or the Ru(II) scaffold, but not the organic inhibitor. These studies highlight that enhanced, and even new, activities can be obtained by the incorporation of metal-containing fragments into existing drugs. In contrast, octahedral metal complexes that are not cytotoxic are promising agents for the photocontrol of enzyme inhibition (P450_BM3_, CYP1B1) or the monitoring of enzyme occupancy (CYP3A4).

Some reasonable conclusions that may be drawn from these studies are that CYP binding behavior can be maintained or enhanced when metal-containing components incorporated into CYP inhibitors are hydrophobic or carry a low overall charge. Given the hydrophobic nature of most CYP substrates, this conclusion may have been anticipated but is supported by recent studies. Unfortunately, one drawback is the potential for unanticipated interactions with other systems in the complex biological environment. Alternatively, chemical features can be leveraged to *prevent* binding; these include the addition of components that are polar and carry higher overall charges, as well as the intentional exploitation of size incompatibilities between the enzyme active site and the metal-containing system.

Other reports in the literature revealed additional factors that should be considered in targeting CYPs with metal-containing compounds. It is known that drug-metabolizing CYPs can bind more than one substrate, and some have more than one binding site, so the possibility of engaging and potentially inhibiting the enzyme while another molecule is already in the active site should be considered. This may be a more general feature of CYPs, even those that are considered selective, as an additional binding site was recently identified in CYP17A1 [128]. Moreover, surface binding that can interfere with electron transfer from POR, as discussed in the literature, should be considered.

Finally, while metal-containing compounds appear to be potential inhibitors and inducers of CYPs, the (limited) data available suggests that they are not generally good substrates. This could be due to a number of factors, including steric features that prevent the requisite close approach of a reactive carbon to the iron oxo moiety, or possibly redox-mediated effects. It is known that some metal complexes, such as Co(III/II) sepulchrate, can act as an electron transfer mediator to drive turnover of P450cin [129] and P450_BM3_ [130], thus it is plausible that other metal complexes may impede enzyme turnover by an electron transfer that *inactivates* the iron heme. Alternatively, in an inorganic inhibitor, the coordinated metal could render the organic ligands electron-deficient and thus less reactive to oxidative modification [131]. As metal binding can be expected to make ligands electron-poor, this seems like a promising general characteristic; however, alternative metabolism by aldehyde oxidase and/or xanthine oxidase must be considered, as these enzymes still act on electron-poor aromatic heterocycles. It is also possible that the inorganic component will alter the site of oxidative modification, as has been shown to be the case with the incorporation of electronegative nitrogen atoms in organic systems [132]. Thus, while some drug molecules may be protected from CYP metabolism by binding to metal centers, it is possible that this strategy will not prevent the other mechanisms of metabolism and excretion that plague drug discovery efforts.

## Figures and Tables

**Figure 1 biomolecules-14-00441-f001:**
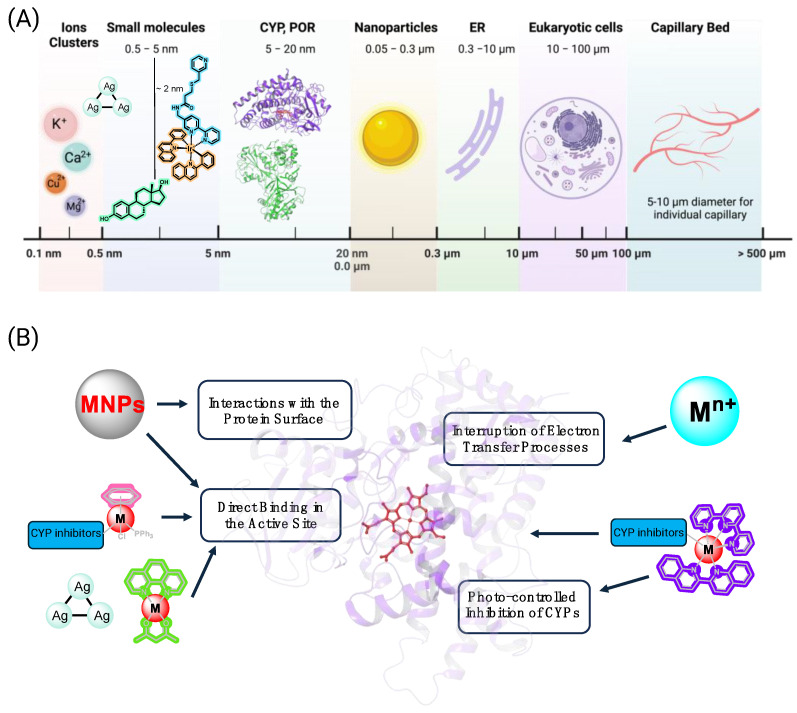
Size parameters, potential interaction sites, and mechanisms of action of inorganic entities that impact Cytochrome P450s. Kinetically labile metal centers, such as alkaline earth ions, undergo rapid ligand exchange with solvent and amino acids, while kinetically inert metal centers in organometallic and coordination compounds are considered stable and do not exchange. (**A**) Relative scales of systems addressed in this review, ranging from ions, clusters, small molecules, proteins, and nanoparticles, to organelles, cells, and vascularized tissues. (**B**) Potential mechanisms of action for disruption of CYP function. Transcriptional regulation is not shown. Image created with Biorender.

**Figure 2 biomolecules-14-00441-f002:**
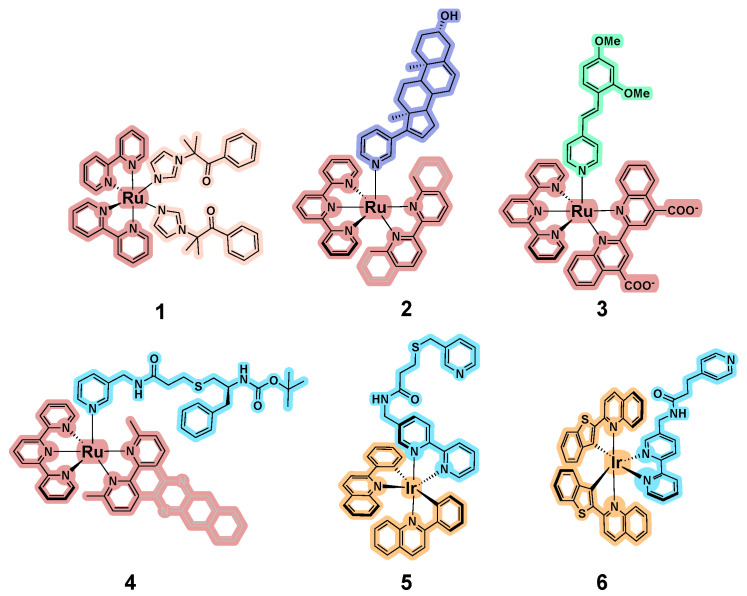
Light-activated inorganic photocages and binders of CYPs. Ru indicates ruthenium and Ir indicates iridium.

**Figure 3 biomolecules-14-00441-f003:**
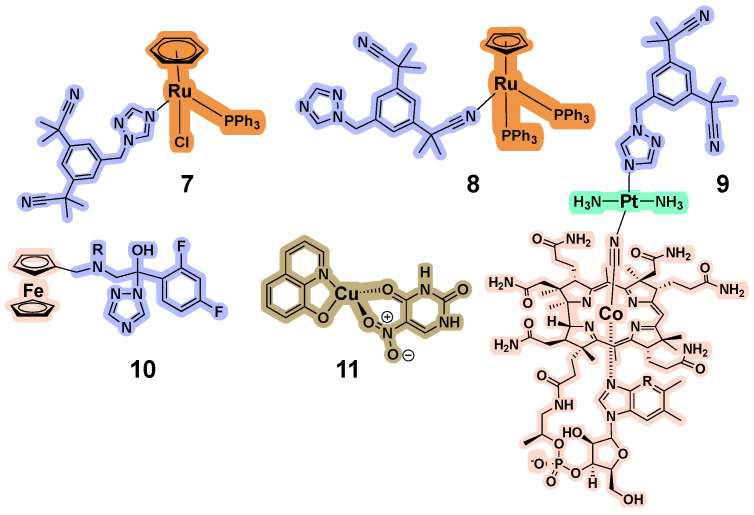
Metal complex CYP inhibitors based on coordinated ligands or overall structure. Ru is ruthenium, Pt is platinum, Co is cobalt, Fe is iron, and Cu is copper.

**Figure 4 biomolecules-14-00441-f004:**
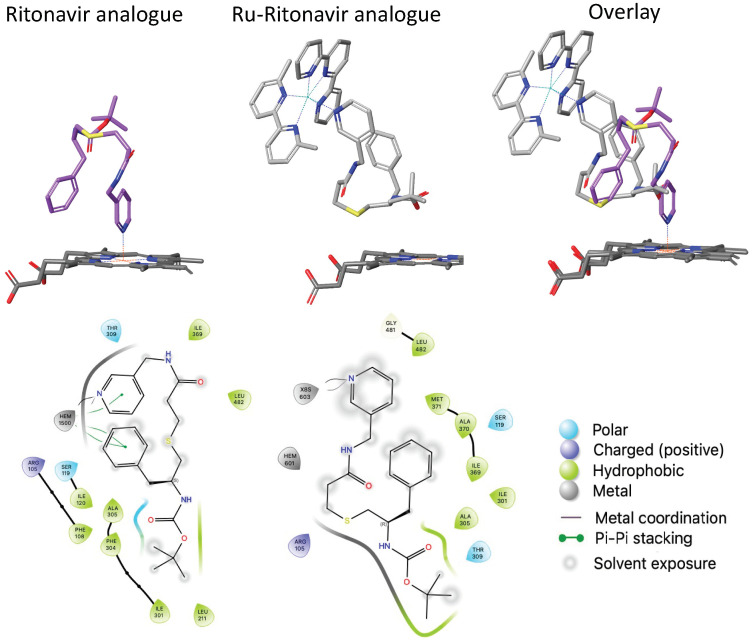
Comparison of CYP3A4 structures with organic and metal-containing inhibitors. Specific hydrophobic interactions are highlighted, along with π–π stacking, polar contacts, and the presence of solvent-exposed portions of the inhibitors. Created with Schrödinger Maestro from PDB 4D78 and 7KS8.

**Table 1 biomolecules-14-00441-t001:** Experimental results of light-activated metal compounds for CYP inhibition.

Compound	CYP	Assay	Protein Binding, Dark	Protein Binding, Light	Parent Ligand Protein Binding,	Dark Inhibition,	Light Inhibition,	Parent Ligand,
			K_d_ (nM)	K_d_ (nM)	K_d_ (nM)	IC_50_ (µM)	IC_50_ (µM)	IC_50_ (µM)
**1**	102A1 (P450BM3)	Protein				6.8	0.05	0.06
**2**	17A1	Protein		89	<100			
**3**	1B1	Cells				1.9	0.0003	0.00031
	1A1	Cells				>30	>30	4.57
	19A1	Cells				>30	18.8	14.4
	phLM	Microsomes				>30	1.28	0.33
**4**	3A4	Protein		340		0.9	2.2	1.54
**5**	3A4	Protein	70			0.25		
**6**	3A4	Protein				2.3		
	3A4	Cells				16		
	1A2	Protein				72		
	2C9	Protein				17

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
