# Peer review of "The Impact of Inorganic Systems and Photoactive Metal Compounds on Cytochrome P450 Enzymes and Metabolism: From Induction to Inhibition"

_biomolecules, 2024, doi:10.3390/biom14040441_

Round 1

Reviewer 1 Report

Comments and Suggestions for Authors

The manuscript presents intriguing points; however, I find it difficult to follow and suggest that it should be rewritten for clarity to the reader. In the first chapter, there is a description of the effect of metal nanoparticles on CYP. It remains unclear whether the data were obtained from cells, microsomes, or purified enzymes. For each referenced study, it is essential to specify the experimental model used for observation. Moreover, it is necessary to consider the ability of nanomaterials to traverse cell membranes and all intracellular processes, which are bypassed when the effect is measured in subcellular preparations and/or pure enzymes.

 The chapter regarding coordination complexes is the least clear. In humans alone, there are 57 genes capable of expressing CYP isoforms, each presenting different pockets and substrates, or varying affinity for the single molecule. Therefore, it should be emphasized when a compound, for example, inhibits CYP102, it also inhibits other CYPs, and what the biological function of these effects may be. Another point that requires clarification is whether the presence of ruthenium itself is an inhibitory factor of CYPs (there are examples in the literature where only the metal is tested) or, as partly described in the manuscript, the presence of ruthenium substantially modifies the physicochemical properties of the drugs under examination and promotes a higher inhibitory capacity against CYPs.

 The chapter regarding induction could be eliminated. Additionally, the title should be changed to avoid misunderstanding; In my opinion better would be "The Impact of Metal and Organometallic Derivatives on Cytochrome P450 Enzymes."

Author Response

The manuscript presents intriguing points; however, I find it difficult to follow and suggest that it should be rewritten for clarity to the reader. In the first chapter, there is a description of the effect of metal nanoparticles on CYP. It remains unclear whether the data were obtained from cells, microsomes, or purified enzymes. For each referenced study, it is essential to specify the experimental model used for observation. Moreover, it is necessary to consider the ability of nanomaterials to traverse cell membranes and all intracellular processes, which are bypassed when the effect is measured in subcellular preparations and/or pure enzymes.

We have added extensive text to address these points, and now each study is fully described.

 The chapter regarding coordination complexes is the least clear. In humans alone, there are 57 genes capable of expressing CYP isoforms, each presenting different pockets and substrates, or varying affinity for the single molecule. Therefore, it should be emphasized when a compound, for example, inhibits CYP102, it also inhibits other CYPs, and what the biological function of these effects may be. Another point that requires clarification is whether the presence of ruthenium itself is an inhibitory factor of CYPs (there are examples in the literature where only the metal is tested) or, as partly described in the manuscript, the presence of ruthenium substantially modifies the physicochemical properties of the drugs under examination and promotes a higher inhibitory capacity against CYPs.

We have added extensive text to address these points, and also discuss which metals are labile (and can be considered to bind to amino acids as free ions) vs. inert, and thus retain their ligands.

 The chapter regarding induction could be eliminated. Additionally, the title should be changed to avoid misunderstanding; In my opinion better would be "The Impact of Metal and Organometallic Derivatives on Cytochrome P450 Enzymes."

We have significantly added to the Induction chapter, and have also modified the title (which was in line with another reviewer’s suggestion).

Reviewer 2 Report

Comments and Suggestions for Authors

Review report

Manuscript Biomolecules-2863338

The manuscript of a review entitled “The Impact of Inorganic Systems on Cytochrome P450 Enzymes and Metabolism: From Induction to Inhibition” submitted by Dmytro Havrylyuk et al. is devoted to the interesting and worth exploring issues. Authors present the survey of the literature on the influence of metal ions, clusters, nanoparticles, complexes and organometallic compounds on the expression and activities of cytochrome P450 enzymes. Recent developments in this field and limitations of the structural studies of enzyme interactions with compounds containing metal atoms are discussed.

To improve the quality of this valuable manuscript I would propose some corrections:

1.     The role of cytochromes P450 as target enzymes could be presented more broadly.

2.     Some changes of the words are suggested:

line 20: breakdown to (more general) biotransformation

line 162: synergetic to synergistic

line 243: in zebrafish embryos

line 263: ketoconazole to ketoconazole

line 301: manor to manner

line 332: Structural studies on interactions of inorganic compounds with CYPs.

line 375: a tether to a linker (more frequently used)

line 470: CYP1B1 protein to CYP1B1 activity

line 478: BM3 to CYP102A1

3.     The authors could consider to replace the term “inorganic compounds”, as a bit confusing in case of metal complexes and organometallic compounds, with “metal containing compounds”.

4.     Some sentences seem to be unclear: lines 147-140; 275-277; 225-227 (probably compared to complex 5?); 427-429. Line 471: Ir(III) complex with pyridine as a ligand?

5.     Figure 3 is not easy to interpret, moreover, the origin (citation) of this figure could be indicated and the caption to the figure could be completed with more details with regard to ligand-enzyme interactions.

6.     In the section 3.6.” Structural studies on inorganic compounds and CYPs” citations are lacking in some paragraphes: 352-361; 362-479; 394-400.

Line 336: It would be better: Meggers and coworkers, or Bregman and coworkers, or Meggers research group

7.     I would advise to explain the abbreviations in the text, for example, tpy (tpy = 2,2:6,2-terpyridine), bpy (2,2’-bipyridine), PI, phototoxicity index

8.     The references could be carefully checked with respect to editor requirements. In references the positions 24 and 25 are repeated. Position 70 is wrongly cited, It should be:

Czekaj P, Skowronek R. Transcription Factors Potentially Involved in Regulation of Cytochrome P450 Gene Expression [Internet], edited by James Paxton. Topics on Drug Metabolism. InTech; 2012. Available from: http://dx.doi.org/10.5772/27817

Author Response

The manuscript of a review entitled “The Impact of Inorganic Systems on Cytochrome P450 Enzymes and Metabolism: From Induction to Inhibition” submitted by Dmytro Havrylyuk et al. is devoted to the interesting and worth exploring issues. Authors present the survey of the literature on the influence of metal ions, clusters, nanoparticles, complexes and organometallic compounds on the expression and activities of cytochrome P450 enzymes. Recent developments in this field and limitations of the structural studies of enzyme interactions with compounds containing metal atoms are discussed.

To improve the quality of this valuable manuscript I would propose some corrections:

  1. The role of cytochromes P450 as target enzymes could be presented more broadly.

We appreciate the suggestion and added two paragraphs to the Introduction.

  1. Some changes of the words are suggested:

line 20: breakdown to (more general) biotransformation

line 162: synergetic to synergistic

line 243: in zebrafish embryos

line 263: ketoconazole to ketoconazole

line 301: manor to manner

line 332: Structural studies on interactions of inorganic compounds with CYPs.

line 375: a tether to a linker (more frequently used)

line 470: CYP1B1 protein to CYP1B1 activity

line 478: BM3 to CYP102A1

We made all these changes.

  1. The authors could consider to replace the term “inorganic compounds”, as a bit confusing in case of metal complexes and organometallic compounds, with “metal containing compounds”.

Done.

  1. Some sentences seem to be unclear: lines 147-140; 275-277; 225-227 (probably compared to complex 5?); 427-429. Line 471: Ir(III) complex with pyridine as a ligand?

These have been edited and expanded.

  1. Figure 3 is not easy to interpret, moreover, the origin (citation) of this figure could be indicated and the caption to the figure could be completed with more details with regard to ligand-enzyme interactions.

Done (there is no citation, as we performed the structural analysis ourselves, which is now discussed in detail in the following paragraph).

  1. In the section 3.6.” Structural studies on inorganic compounds and CYPs” citations are lacking in some paragraphes: 352-361; 362-479; 394-400.

Citations have been added.

Line 336: It would be better: Meggers and coworkers, or Bregman and coworkers, or Meggers research group

Done

  1. I would advise to explain the abbreviations in the text, for example, tpy (tpy = 2,2:6,2-terpyridine), bpy (2,2’-bipyridine), PI, phototoxicity index

Done

  1. The references could be carefully checked with respect to editor requirements. In references the positions 24 and 25 are repeated. Position 70 is wrongly cited, It should be:

Czekaj P, Skowronek R. Transcription Factors Potentially Involved in Regulation of Cytochrome P450 Gene Expression [Internet], edited by James Paxton. Topics on Drug Metabolism. InTech; 2012. Available from: http://dx.doi.org/10.5772/27817

We thank the reviewer for bringing this to our attention!

Reviewer 3 Report

Comments and Suggestions for Authors

The Impact of Inorganic Systems on Cytochrome P450 Enzymes and

Metabolism: From Induction to Inhibition

Biomolecules 2863338

General comments

This is a nicely written and interesting minireview on the topic of inorganic compound interactions with CYPs. Some additional guidance for the reader and overall summarization of the topics discussed would help with understanding at a glance which topics are covered and what the mechanisms of interaction are. I have no major criticisms but include a number of minor comments for the further refinement of the manuscript below.

Minor comments

(1) An overview figure showing the mechanisms of interaction of the inorganic compounds with CYPs would be very helpful. This would help the reader understand better the content since the sections are mainly organized by chemistry rather than mechanism. Such a figure could help us to see clearly the multiple points of potential interaction: from required metal binding for optimal protein folding, potential interruption of electron transfer processes, direct binding in the active site and release of organic molecules from complexes that then inhibit the CYP. This could enable readers newer to the field to grasp the different concepts described in the review more easily.

(2) In a similar manner, an overview table showing the main classes of compounds, their interaction mechanisms and key examples would be helpful to the article.

(3) Consider a modification to the title to enable readers interested in a review on photoactivated CYP inhibitors to find this work. These molecules currently occupy a large part of the review but do not seem to be represented in the title. Conversely, there is little on induction in the review although induction is included in the title. [Probably induction could be left out of the title.]

(4) Treatment of chelated CYP inhibitor molecules and IC50 values in the light and dark forms quite a section of the review. At present these are treated as if either the inhibitory moiety is totally chelated or as if it is all photodissociated. Some more nuanced discussion about the binding affinities for the chelates, the complex dynamic equilibria involved, the extent of photodissociation and comparisons with the organic molecule used independently as an inhibitor would be welcome. Maybe this could come as introductory text to a section. Along these lines, the discussion in section 3.6 just below figure 3 was highly interesting.

(5) There is quite a lot of discussion about CYP3A4, one of the most important CYP enzymes. It would be interesting to see some of the historical structure-activity work performed on the enzyme to study amino acid active site interactions and active site volume constraints also brought in to complement the discussion sections in lines 362-373. Example references include:

https://doi.org/10.1124/dmd.30.4.452;
DOI: 10.1021/bi010758a

Author Response

General comments

This is a nicely written and interesting minireview on the topic of inorganic compound interactions with CYPs. Some additional guidance for the reader and overall summarization of the topics discussed would help with understanding at a glance which topics are covered and what the mechanisms of interaction are. I have no major criticisms but include a number of minor comments for the further refinement of the manuscript below.

Thank you!

Minor comments

  • An overview figure showing the mechanisms of interaction of the inorganic compounds with CYPs would be very helpful. This would help the reader understand better the content since the sections are mainly organized by chemistry rather than mechanism. Such a figure could help us to see clearly the multiple points of potential interaction: from required metal binding for optimal protein folding, potential interruption of electron transfer processes, direct binding in the active site and release of organic molecules from complexes that then inhibit the CYP. This could enable readers newer to the field to grasp the different concepts described in the review more easily.

This was a fantastic suggestion, and we have created a new Figure (Figure 1) that we hope will orient the reader to the different mechanisms of interaction as well as the related topic of the variation in size scale for the inorganic systems being discussed (from individual ions to nanoparticles).

  • In a similar manner, an overview table showing the main classes of compounds, their interaction mechanisms and key examples would be helpful to the article.

We have added Table 1, although this deals more with specific features of specific inhibitors.

  • Consider a modification to the title to enable readers interested in a review on photoactivated CYP inhibitors to find this work. These molecules currently occupy a large part of the review but do not seem to be represented in the title. Conversely, there is little on induction in the review although induction is included in the title. [Probably induction could be left out of the title.]

We changed the title as recommended, and added significantly to the section of induction.

(4) Treatment of chelated CYP inhibitor molecules and IC50 values in the light and dark forms quite a section of the review. At present these are treated as if either the inhibitory moiety is totally chelated or as if it is all photodissociated. Some more nuanced discussion about the binding affinities for the chelates, the complex dynamic equilibria involved, the extent of photodissociation and comparisons with the organic molecule used independently as an inhibitor would be welcome. Maybe this could come as introductory text to a section. Along these lines, the discussion in section 3.6 just below figure 3 was highly interesting.

We have added extensive discussion to address these issues, and explain kinetically inert vs. labile metal complexes and equilibria in biological media.

(5) There is quite a lot of discussion about CYP3A4, one of the most important CYP enzymes. It would be interesting to see some of the historical structure-activity work performed on the enzyme to study amino acid active site interactions and active site volume constraints also brought in to complement the discussion sections in lines 362-373. Example references include:https://doi.org/10.1124/dmd.30.4.452; DOI: 10.1021/bi010758a

We agree with the helpful recommendation, have added two paragraphs on this topic and point the reader to illustrative literature.

Reviewer 4 Report

Comments and Suggestions for Authors

The manuscript, submitted for publication in Biomolecules, by Dmytro Havrylyuk et al. entitled: "The impact of inorganic systems on cytochrome P450 enzymes and metabolism: from induction to inhibition" reviews, to my opinion for the first time, the ‘inorganic life’ of P450 enzymes, i.e. how these hemoproteins are modulated by metals and metal complexes (both organometallic and coordination complexes). This review comprises 8 main paragraphs covering from P450 inhibition by various metal complexes to the structure of P450 with bound metal complexes and the induction of P450 expression by metal complexes. I learned a lot of new things and I have found that this paper will be a source of useful information for the readership of Biomolecules und also for people working on P450 enzymes, up to my knowledge, in particular for the valuable information on selective P450 inhibition by metal complexes. To my opinion, this comprehensive review, with 77 bibliographical references, thus merits to be published in Biomolecules provide some comments and questions I have be answered by the authors.

Major comments

1. Page 1. Lines 41-43. Are they any structural or experimental explanation of this intriguing differential behavior toward human liver P450s? Same question for the differential behavior observed with AG clusters.

2. Page 3. Line 97. The sentence part: “… with effects observed over long distances between the metal and the heme …;” seems to be in full contradiction with the previous sentence (at line 96) stating: “… ions present in the active site ..”. Could the authors comment on that?

3. Page 3. Concerning the wavelength (450 nm) which is used to irradiate the metal complexes (lines 125-127). What about the two flavins that are bound to CYP102A1 that will absorb most of this light intensity at this exact same wavelength? Would not it introduce some bias in this experiment? Could the authors comment on that?

4. Page 3. Line 123. We read: “… CYP102A1 was chosen for the studies, as it is a model for promiscuous drug metabolizing hepatic CYPs”. I am not sure that CYP102A1 is “that” a good model for hepatic drug metabolizing P450s since it has to be largely optimized for a good production of human drug metabolites (See: Int J Mol Sci 2012 13(12), 15901-15924).

5. Page 5. Line 212. We read: “… CYP3A4 is a promiscuous enzyme with a large volume active site cavity (1400 Å3) …” As such it is untrue. It is known since a long time that the catalytic cavity of CYP3A4 may change in volume more than a thousand Å3 depending on the nature of the bound ligand (See: Teixeira VH et al. BBA Proteins and Proteomics 2010, 1804(10), 2036-2045).

6. Page 7. Sentence, lines 280-283. This PAH (i.e. dimethylphenanthroline) is a ligand (and even a substrate) of human CYP1A1. This fact contradicts the previous sentence that states: “… metal complexes that do not contain CYP inhibitors …” since a substrate for an enzyme is an inhibitor of another substrate of the same enzyme. Could the authors comment on that?

Minor

1. Page 2. A scheme showing the arrangements of silver ions in AG2 and AG3 clusters would be most welcomed. What are the sizes of these clusters?

2. Page 2. Line 68. There is an error here. This sentence should read: “The impact of (…) has also been investigated …”

3. Page 3. Sentence lines 121-123. Please introduce a reference for this statement.

4. Please change Kd for the more exact constant writing KD (with an upper-case D).

5. Page 4. Fig. 1 legend. Please wirte in the legend that Ru and Ir stands for ruthenium and iridium, respectively.

6. Page 4, line 218. There is a typo here. 170 nm should read 170 nM.

7. Page 5. Lines 240-243. A reference to this statement is missing.

8. Page 5. Line 263. There is a typo here. Ketoconozole should read ketoconazole.

9. Page 7. Line 294. To which molecule each of these two Ks correspond? Please indicate them in the text.

10. Page 8. Line 362. Please precise: “parts of the CYP structure …”.

11. Page 9. Lines 376-377. Please provide the PDB entry.

Author Response

Reviewer 4

The manuscript, submitted for publication in Biomolecules, by Dmytro Havrylyuk et al. entitled: "The impact of inorganic systems on cytochrome P450 enzymes and metabolism: from induction to inhibition" reviews, to my opinion for the first time, the ‘inorganic life’ of P450 enzymes, i.e. how these hemoproteins are modulated by metals and metal complexes (both organometallic and coordination complexes). This review comprises 8 main paragraphs covering from P450 inhibition by various metal complexes to the structure of P450 with bound metal complexes and the induction of P450 expression by metal complexes. I learned a lot of new things and I have found that this paper will be a source of useful information for the readership of Biomolecules und also for people working on P450 enzymes, up to my knowledge, in particular for the valuable information on selective P450 inhibition by metal complexes. To my opinion, this comprehensive review, with 77 bibliographical references, thus merits to be published in Biomolecules provide some comments and questions I have be answered by the authors.

Major comments

  1. Page 1. Lines 41-43. Are they any structural or experimental explanation of this intriguing differential behavior toward human liver P450s? Same question for the differential behavior observed with AG clusters.

We were also very interested in this question, but could find no information in the literature, so we can’t provide any rational hypothesis for it.

  1. Page 3. Line 97. The sentence part: “… with effects observed over long distances between the metal and the heme …;” seems to be in full contradiction with the previous sentence (at line 96) stating: “… ions present in the active site ..”. Could the authors comment on that?

We have clarified and expanded this section to make clear that the “long distances” are up to 20 A from the heme, and the number and nature of the ions matters for the impact on the activity.

  1. Page 3. Concerning the wavelength (450 nm) which is used to irradiate the metal complexes (lines 125-127). What about the two flavins that are bound to CYP102A1 that will absorb most of this light intensity at this exact same wavelength? Would not it introduce some bias in this experiment? Could the authors comment on that?

This is an excellent point! We added a discussion of the truncation of the CYP102A1 used in the studies, which removes the flavin binding domain and thus prevents light-induced transitions in these cofactors.

  1. Page 3. Line 123. We read: “… CYP102A1 was chosen for the studies, as it is a model for promiscuous drug metabolizing hepatic CYPs”. I am not sure that CYP102A1 is “that” a good model for hepatic drug metabolizing P450s since it has to be largely optimized for a good production of human drug metabolites (See: Int J Mol Sci 2012 13(12), 15901-15924).

We have clarified the point, including the fact that we used an engineered variant (which binds and performs reactions on a number of drugs) and convenience factors, such as the fact that the enzyme is soluble and easy to express, allowing for biophysical studies.

  1. Page 5. Line 212. We read: “… CYP3A4 is a promiscuous enzyme with a large volume active site cavity (1400 Å3) …” As such it is untrue. It is known since a long time that the catalytic cavity of CYP3A4 may change in volume more than a thousand Å3depending on the nature of the bound ligand (See: Teixeira VH et al. BBA Proteins and Proteomics 2010, 1804(10), 2036-2045).

We have added this important point to the discussion in two places to highlight the plasticity of the active site.

  1. Page 7. Sentence, lines 280-283. This PAH (i.e. dimethylphenanthroline) is a ligand (and even a substrate) of human CYP1A1. This fact contradicts the previous sentence that states: “… metal complexes that do not contain CYP inhibitors …” since a substrate for an enzyme is an inhibitor of another substrate of the same enzyme. Could the authors comment on that?

A sentence has been added to address this.

Minor

  1. Page 2. A scheme showing the arrangements of silver ions in AG2 and AG3 clusters would be most welcomed. What are the sizes of these clusters? We agree! We added this in the new Figure 1.
  2. Page 2. Line 68. There is an error here. This sentence should read: “The impact of (…) has also been investigated …” Thank you!
  3. Page 3. Sentence lines 121-123. Please introduce a reference for this statement. Thank you
  4. Please change Kdfor the more exact constant writing KD(with an upper-case D). We have chosen to keep the lower case letter based on the usage in textbooks such as those of Robert Copeland.
  5. Page 4. Fig. 1 legend. Please wirte in the legend that Ru and Ir stands for ruthenium and iridium, respectively. Done.
  6. Page 4, line 218. There is a typo here. 170 nm should read 170 nM. Thank you!
  7. Page 5. Lines 240-243. A reference to this statement is missing. Thank you!
  8. Page 5. Line 263. There is a typo here. Ketoconozole should read ketoconazole. Thank you!
  9. Page 7. Line 294. To which molecule each of these two Ks correspond? Please indicate them in the text. Thank you!
  10. Page 8. Line 362. Please precise: “parts of the CYP structure …”. This is now corrected.
  11. Page 9. Lines 376-377. Please provide the PDB entry. Thank you!

Round 2

Reviewer 1 Report

Comments and Suggestions for Authors

The authors have improved the text, which now presents itself more clearly to readers. in this version the manuscript is worthy of publication